# Multigene panel next generation sequencing in metastatic colorectal cancer in an Australian population

Udit Nindra[1,2,3,4,5]*, Abhijit Pal[1,3,4], Vivienne Lea[6], Stephanie Hui-Su Lim[2,4,5], Kate Wilkinson[1], Ray Asghari[3], Tara L. Roberts[4,5,7], Therese M. Becker[4,5,7], Mahtab Farzin[6], Tristan Rutland[5,6], Mark Lee[7], Scott MacKenzie[5], Weng Ng[1,4,5,7], Bin Wang[6], C. Soon Lee[3,5,6,7‡], Wei Chua[1,4,5,7‡]

1 Department of Medical Oncology, Liverpool Hospital, Liverpool, New South Wales, Australia, 2 Department of Medical Oncology, Macarthur Cancer Therapy Centre, Campbelltown Hospital, Campbelltown, New South Wales, Australia, 3 Department of Medical Oncology, Bankstown-Lidcombe Hospital, Bankstown, New South Wales, Australia, 4 Ingham Institute for Applied Medical Research, Liverpool, Australia, 5 School of Medicine, Western Sydney University, Sydney, Australia, 6 Department of Anatomical Pathology, Liverpool Hospital, Liverpool, Sydney, Australia, 7 South Western Clinical School, University of New South Wales, Sydney, Australia

‡ CSL and WC are joint senior authors on this work.
* udit.nindra@health.nsw.gov.au

**Data Availability Statement:** All relevant data are within the paper and its Supporting information files.

## Abstract

### Background

Next generation sequencing (NGS) is increasingly used in standard clinical practice to identify patients with potentially actionable mutations. Stratification of NGS mutation tiers is currently based on the European Society of Medical Oncology (ESMO) Scale for Clinical Actionability of Molecular Targets (ESCAT[E]) Tier I–V & X. Allele frequency is also increasingly recognised as an important prognostic tool in advanced cancer. The aim of this study was to determine the genomic mutations in metastatic colorectal cancer (CRC) in an Australian multicultural population and their influence on survival outcomes.

### Methods

Next generation sequencing with the 50-gene panel Oncomine Precision Assay™ was used on 180 CRC tissue samples obtained across six Sydney hospitals between June 2021 and March 2022.

### Results

From 180 samples, 147 (82%) had at least one gene mutation identified with 68 (38%) having two or more concurrent mutations. Tier I variants included *RAS* wild-type [EI] in 73 (41%) and *BRAF V600E* [EIA] in 27 (15%). Non-tier I variants include 2 (1%) *ERBB2* amplification [EIIB], 26 (15%) *PIK3CA* hotspot mutations [EIIIA] and 9 (5%) *MET* focal amplifications [EIIIA]. NGS testing revealed an additional 22% of cases with Tier II & III mutations. 43% of patients also presented with potentially actionable Tier III & IV mutations. Patients with concurrent *TP53* and *RAS* mutations had significantly reduced overall survival (6.1

**Funding:** The author(s) received no specific funding for this work.

**Competing interests:** The authors have declared that no competing interests exist.

months versus 21.1 months, p <0.01). High *KRAS* allele frequency, as defined by those with over 20% variant allele frequency (VAF), also demonstrated reduced overall survival (12.1 months versus 42.9 months, p = 0.04).

## Conclusions

In addition to identifying patients with genomic alterations suitable for clinically proven standard of care therapeutic options, the 50 gene NGS panel has significant potential in identifying potentially actionable non-tier 1 mutations and therefore may become future standard clinical practice.

## 1. Introduction

Colorectal Cancer (CRC) continues to remain a significant cause of cancer death worldwide [1]. Initial treatment of metastatic CRC has been largely reliant on chemotherapy and targeted therapy but recent advances in technology have introduced the role of genomic testing to identify druggable targets, either established or currently being tested in clinical trial settings. One example of actionable and targetable mutations that has influenced treatment of CRC is the BRAF V600E mutation. Recent data has shown efficacy of mutant BRAF ATP-competitive RAF kinase inhibitors, such as encorafenib, in combination with cetuximab, a monoclonal antibody targeting the epidermal growth factor receptor (EGFR) receptor, for the treatment of BRAF V600E mutated CRC [2]. The ability to detect mutations in tumours not only provides clinicians and researchers with possible therapeutic targets but also provides an understanding of potential resistance mechanisms to particular treatments. For example, KRAS mutations in CRC are known to cause resistance to EGFR monoclonal antibody therapies such as cetuximab and panitumumab [3, 4]. Additionally, the presence of certain mutations can be associated with higher risk histological features at diagnosis. For example, PIK3CA hotspot mutations are reported to occur in approximately 16% of CRC (5). These mutations have been shown to correlate with poor histological grade, late clinical stage at presentation, worse prognosis and first line chemotherapy resistance [5].

Next Generation Sequencing (NGS) provides rapid testing of multiple genetic molecular variants to inform clinicians of therapeutic options. Targeted NGS technology relies on known mutational molecular variants with the ability to streamline detection for a panel of variants of clinical interest. NGS technology can identify hotspot mutations and copy number variant mutations in tumour DNA as well as fusion transcript variants in tumour RNA. This can lead to the ability to not only identify specific variant mutations of interest but also the variant allele frequency. Allele frequency is currently not used in clinical practice to influence treatment decisions but early evidence, especially in the *EGFR* mutated NSCLC setting, suggests it could play a role in predicting therapeutic efficacy [6, 7]. NGS testing in patients with stage IV CRC has been increasingly utilised to help determine the genetic profile of tumours and possible resistance mechanisms through concomitant mutations. NGS using the Oncomine Precision Assay™ provides a means of testing for mutations across 50 genes, namely: *KRAS, NRTK, BRAF, NRAS, ERBB2, RET, MET, ROS1, PIK3CA, NRG1, AKT1, AKT2, AKT3, CDK4, CTNNB1, ERBB3, FGFR2, FGFR3, FLT3, GNA11, GNAQ, GNAS, HRAS, IDH1, IDH2, MAP2K1, MAP2K2, PDGFRA, PTEN, SMO, TP53, EGFR, ALK, AR, ARAF, RAF1, MTOR, KIT, FGFR, ERBB4, ESR1, FGFR1, CDKN2A* and *CHEK2*. Although not all these mutations currently guide therapeutic options, emerging understanding of the roles of these genes and

novel drug development may provide further information into prognostication of patients, treatment options, or predict therapy resistance.

Currently, there is no standardised method for classifying mutations identified with NGS technology into tiers of importance. The European Society of Medical Oncology (ESMO) Precision Medicine Working Group has proposed a set of guidelines, defining mutations from Tier 1 to Tier X actionability which are summarised in Table 1 [8]. Here we used the Oncomine Precision Assay™ for sequencing of stage IV CRC from large patient numbers to obtain frequencies of mutations in CRC for our real-world, multicultural Australian patient cohort, as stratified by the ESMO Precision Medicine Working Group.

## 2. Methods

### 2.1 NGS mutation analysis

Next generation sequencing was performed with the Ion Torrent Genexus integrated next generation sequencer using the Oncomine Precision Assay™, which is 50-gene NGS panel that was introduced to our laboratory in June 2021. Only DNA hotspot mutations and copy number variations (CNVs) were assessed. RNA sequencing for fusions was not performed. The Ion Torrent Genexus System fully automates the specimen-to-report workflow. It automates all steps of the targeted NGS workflow starting from extraction, purification and quantification of nucleic acid. The Genexus Purification System replaces sample prep by extracting and quantifying nucleic acids. This is followed by automation of the library preparation (including DNA synthesis), template preparation and sequencing. Lastly, primary data analysis is also automated and variant reporting for DNA applications are performed using the Oncomine™ Reporter software, a Thermo Fisher supplied bioinformatics analytic tools. The reported results from the 50-gene Oncomine™ Precision Assay are filtered to provide a summary of the variants and copy number variation as well as the variant allele frequency. Historically, NGS testing using Oncomine panels from surgical samples has a reported success rate of approximately 80% for certain solid organ malignancies [9]. However, this accuracy is variable depending on the NGS system and software used.

### 2.2 Data collection and correlation

This is a retrospective observational cohort study utilising CRC patient NGS and clinical records. We included all patients with histologically confirmed metastatic CRC that had

**Table 1. ESMO tiers of mutational actionability.**

| Tier | Definition | Evidence required |
|---|---|---|
| I | Ready for routine clinical use. Access to treatment should be considered standard of care. | I-A: Prospective RCT demonstrating improvement of survival end point<br>I-B: Prospective non-randomised trials demonstrating meaningful benefit |
| II | Investigational mutation. The treatment is considered preferrable in context of evidence collected | Retrospective studies demonstrate a clinically meaningful survival benefit or prospective studies demonstrate increase response without survival data available |
| III-IV | Hypothetical Target | Clinical benefit seen in patients but not in the same tumour stream. |
| V | Mutation is associated with objective response but without clinically meaningful benefit | Prospective studies show that targeted therapy is associated with objective responses but not improved survival |
| X | Lack of evidence for actionability | No evidence. The genomic alteration is therapeutically actionable. |

Oncomine™ NGS testing between June 2021 and March 2022. In total, records from 180 de-identified patients with CRC were available from 5 hospitals across greater Sydney, Australia. We collected patient demographic data including age at diagnosis, sex and ethnicity (as defined by country of birth) in addition to NGS mutation reports. Data regarding tumour differentiation and mismatch repair (MMR) status was collected where possible. The project was approved by the Sydney South West Local Health District Human Research Ethics Committee (2019/ETH04187). Survival outcomes were also collected with data censored as of the end of February 2023. We stratified identified mutations according to the ESMO guidelines as documented in Table 1. We correlated mutation frequency with clinicopathological variables and survival outcomes using cox-proportional hazard ratios and Kaplan-Meier curve analyses A p-value of 0.05 was defined as statistically significant. All data analysis was done using R 4.1.1.

## 3. Results

### 3.1 Clinicopathological characteristics and survival outcomes

One hundred and eighty patients with CRC were included in the analysis. The median age was 67 with 59% of the patients being male. All patients had histologically confirmed adenocarcinoma. Sixty-nine (38%) had moderately differentiated disease, with majority being unknown at the time of testing (50%). 86 (47%) had mismatch repair (MMR) testing completed with 7 (4%) demonstrating deficient MMR status. Baseline patient demographics are summarised in Table 2.

**3.1.1 Genomic profiling for actionable mutations and potentially actionable targets.** Out of 180 patients, 147 (82%) had at least one gene mutation identified via NGS with 68 (38%) having two or more mutations identified. Based on stratification of NGS mutation tiers

**Table 2. Patient demographics.**

| N | 180 |
|---|---|
| Age (median, range) [years] | 67 (22–89) |
| Sex | |
| Male | 104 (58%) |
| Female | 76 (42%) |
| Region of Birth | |
| Australia / New Zealand | 112 (62%) |
| Pacific Islands | 2 (1%) |
| South East Asia | 23 (13%) |
| Eastern Europe | 28 (16%) |
| Middle East | 11 (6%) |
| Other | 3 (2%) |
| Unknown | 1 (1%) |
| Differentiation | |
| Well differentiated | 5 (3%) |
| Moderately differentiated | 69 (38%) |
| Poorly differentiated | 16 (9%) |
| Unknown | 90 (50%) |
| Mismatch Repair Status | |
| Proficient | 79 (44%) |
| Deficient | 7 (4%) |
| Unknown | 93 (52%) |

**Table 3. Mutations identified using next generation sequencing as stratified by the European society of medical oncology scale for clinical actionability of molecular targets guidelines.**

| Mutation | Tier | Frequency |
|---|---|---|
| All RAS / BRAF WT | 1 | 73 (41%) |
| BRAF V600E | 1 | 27 (15%) |
| ERBB2 Amplifications | 2 | 2 (1%) |
| MET Focal Amplifications | 3 | 9 (5%) |
| PIK3CA | 3 | 26 (15%) |
| AKT1 | 3 | 2 (1%) |

as per the ESMO Precision Medicine Working Group (9), 105 (58%) patients had Tier 1 mutations, 59 (33%) of which were patients with tier 1 mutations alone. Overall, 33 (18%) of patients had no mutation identified. Tier I variants included RAS wild-type [EI] in 73 (41%) and BRAF V600E [EIA] in 27 (15%). Non-tier I included 2 (1%) ERBB2 amplification [EIIB], 26 (15%) PIK3CA hotspot mutations [EIIIA] and 9 (5%) MET focal amplifications [EIIIA]. These results are summarised in in Table 3 and Fig 1.

In our cohort, 71 (43%) had at least one of 22 potentially actionable Tier 3–4 mutations. We classified these 27 mutations as Tier 3–4 as there are currently active early phase clinical trials being run to investigate the therapeutic role of targeted agents against these mutations in either CRC or any solid organ malignancy. These 22 genes are summarised in Table 4. We also classified 12 genes as Tier X as these have no current published evidence suggesting that these genes play a role in tumourigenesis, treatment efficacy or resistance. Furthermore, there were no ongoing trials investigating the roles of these genes in CRC or any solid organ malignancy. These were *AR*, *CDKN2A*, *ARAF*, *RAF1*, *MTOR*, *KIT*, *FGFR4*, *ERBB4*, *ERS1*, *FGFR1*, *GNAS* and *CHEK2*.

**3.1.2 Concurrent mutations and their impact on overall survival.** 68 (38%) patients had 2 or more concurrent mutations and these have been summarised in Fig 2. Patients with RAS mutations had a median overall survival of 16.9 months from time of metastatic disease diagnosis. In patients with *RAS* mutated CRC, concurrent presence of *TP53* mutations conferred significantly reduced overall survival (6.1 months versus 21.1 months, HR 4.2, 95% CI 1.7–10.6, p <0.01) as demonstrated in Fig 2. In patients with RAS mutated CRC, concurrent presence of *PIK3CA* mutation did not influence survival.

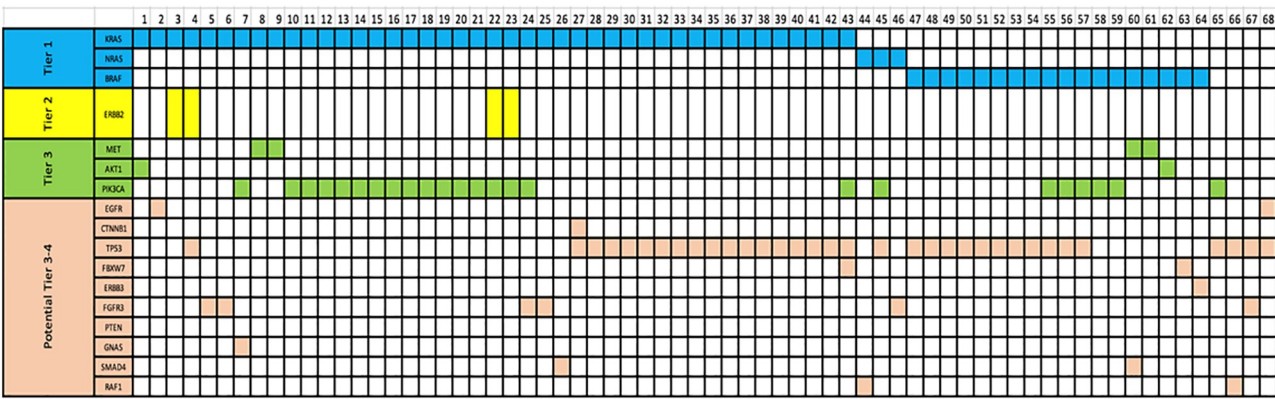

**Fig 1. Distribution of concurrent mutations identified by next generation sequencing in colorectal cancer.** Individual patients with concurrent mutations in each column. Tiers 1–3 as outlined by the European Society of Medical Oncology Scale for Clinical Actionability of Molecular Targets.

**Table 4. Potentially actionable Tier 3–4 mutations.**

| Mutation | Frequency (CRC) | CRC specific clinical trial | Any solid organ malignancy clinical trial |
|---|---|---|---|
| AKT2 | 0 (0%) | NCT01333475 | - |
| AKT3 | 0 (0%) | NCT01333475 | - |
| BRAF (non-V600E) | 2 (2%) | NCT03087071 | - |
| CDK4 | 0 (0%) | - | NCT01037790 |
| CTNNB1 | 3 (2%) | NCT04851119 | NCT04851119 |
| EGFR | 1 (1%) | NCT00326495 | - |
| ERBB3 | 1 (1%) | NCT04479436 | - |
| FGFR2 | 0 (0%) | NCT04096417 | - |
| FGFR3 | 0 (0%) | NCT04096417 | - |
| FLT3 | 0 (0%) | NCT01762293 | - |
| GNA11 | 0 (0%) | - | NCT03947385 |
| GNAQ | 0 (0%) | - | NCT03947385 |
| HRAS | 0 (0%) | NCT04853043 | - |
| IDH1 | 0 (0%) | - | NCT04521686 |
| IDH2 | 0 (0%) | - | NCT04521686 |
| KRAS (non-G12C) | 0 (0%) | NCT03948763 | NCT03948763 |
| MAP2K1 | 1 (1%) | NCT03087071 | NCT04488003 |
| MAP2K2 | 0 (0%) | - | NCT04488003 |
| PDGFRA | 0 (0%) | - | NCT03693326 |
| PTEN | 2 (3%) | - | NCT01458067 |
| SMO | 0 (0%) | - | NCT02002689 |
| TP53 | 60 (33%) | NCT03144804 | NCT00393029 |

**3.1.3 Allele frequency and its impact on survival.** For 44 (24%) patients with *RAS* mutations, allele frequency and survival data was available. High RAS allele frequency, as defined by those with over 20%, demonstrated reduced overall survival (12.1 months versus not reached, HR 2.2, 95%CI 1.3–6.7, p = 0.04). This is demonstrated in Fig 3. RAS allele frequency did not correlate with upfront chemotherapy responsiveness as defined as stable disease or better on first imaging done approximately 3 months after commencement of first line chemotherapy in the metastatic setting. Median progression free survival in high RAS allele frequency patients was 6.8 months versus 10.3 months in low RAS allele frequency patients but this result was not statistically significant (p = 0.22). In 12 patients with BRAF V600E mutations (7%), allele frequency and survival data were available. High BRAF V600E mutation allele frequency, as defined by those with over 15%, trended towards shorter overall survival without statistical significance (9.8 months versus 12.0 months, p = 0.7).

## 3.2 Ethnicity and mutational landscape

Culturally and linguistically diverse (CALD) patients are those who are either born in non-English speaking counties and/or who do not speak English at home [3]. 64/180 (36%) patients in our cohort were CALD background. There was a slightly higher incidence of any tier mutation identified in the CALD cohort compared with non-CALD populations (84% versus 80%) but this result was not statistically significant. RAS mutations were noted in 42% of patients (38% with KRAS mutations and 4% with NRAS mutations). In our cohort, patients from Asia had the highest rate of RAS mutation (52%) compared with any other ethnic subgroup with those of Middle Eastern ethnicity having the lowest rate of RAS mutation (36%) in our cohort. BRAF mutations were noted in 29 (17%) patients with 27/29 having the V600E alteration. The

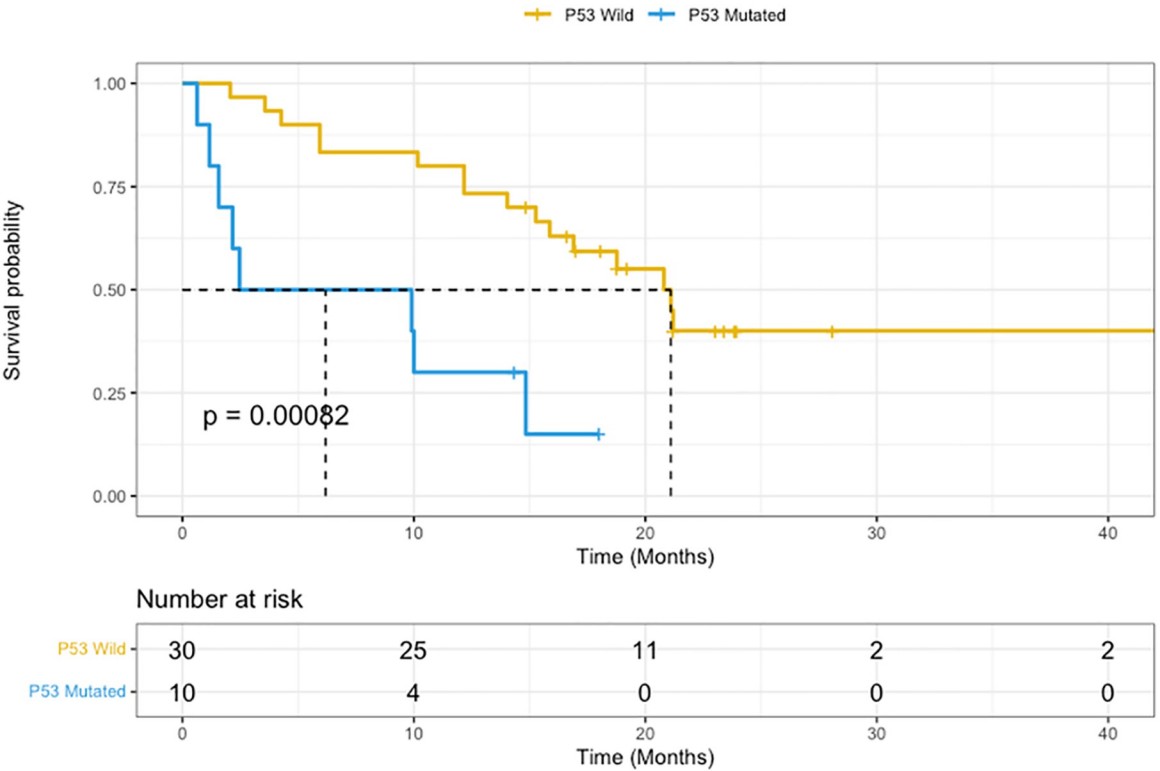

**Fig 2. Overall survival of RAS-mutated metastatic colorectal cancer with and without concurrent P53 mutations.** Kaplan-Meier curve comparing TP53 wildtype (yellow) vs TP53 mutated (blue) patient outcomes. Dotted line represents median (6.1 months versus 21.1 months, HR 4.2, 95% CI 1.7–10.6, p <0.01).

highest incidence of BRAF mutations was noted in the Australian Caucasian population compared with those from Asia or the Middle East (21% versus 18% and 13% respectively). In our cohort, PIK3CA mutations were present in 14% of Australian ethnicity patients, 14% of Eastern European patients, 17% of South East Asian ethnicity patients and 18% of Middle Eastern ethnicity patients. Additionally, one third of patients had some variant of TP53. TP53 mutations appeared to be fairly evenly present across the various ethnic subgroups with 33% of Australian ethnicity patients, 28% of Eastern European patients, 26% of South East Asian ethnicity patients and 45% of Middle Eastern ethnicity patients exhibiting mutations. An overall summary of mutations based on region of birth is presented in Table 5.

## 4. Discussion

NGS testing provides a wealth of information regarding a patient's tumour biology, and informs mechanisms of treatment response and clinical prognosis. Currently, the majority of the information generated is not used in clinical decision making. Our study highlights important hypothesis generating questions including the influence of allele frequency and presence of non-traditional co-mutations on clinical outcomes. Currently, there are no published studies in the CRC space that investigate the impact of RAS allele frequency on clinical outcomes. Friedlaender et al., reported the role of allele frequency on survival outcomes in NSCLC [7]. Their retrospective study demonstrated that allele frequency was an independent prognostic marker of both progression free and overall survival in advanced EGFR-mutated

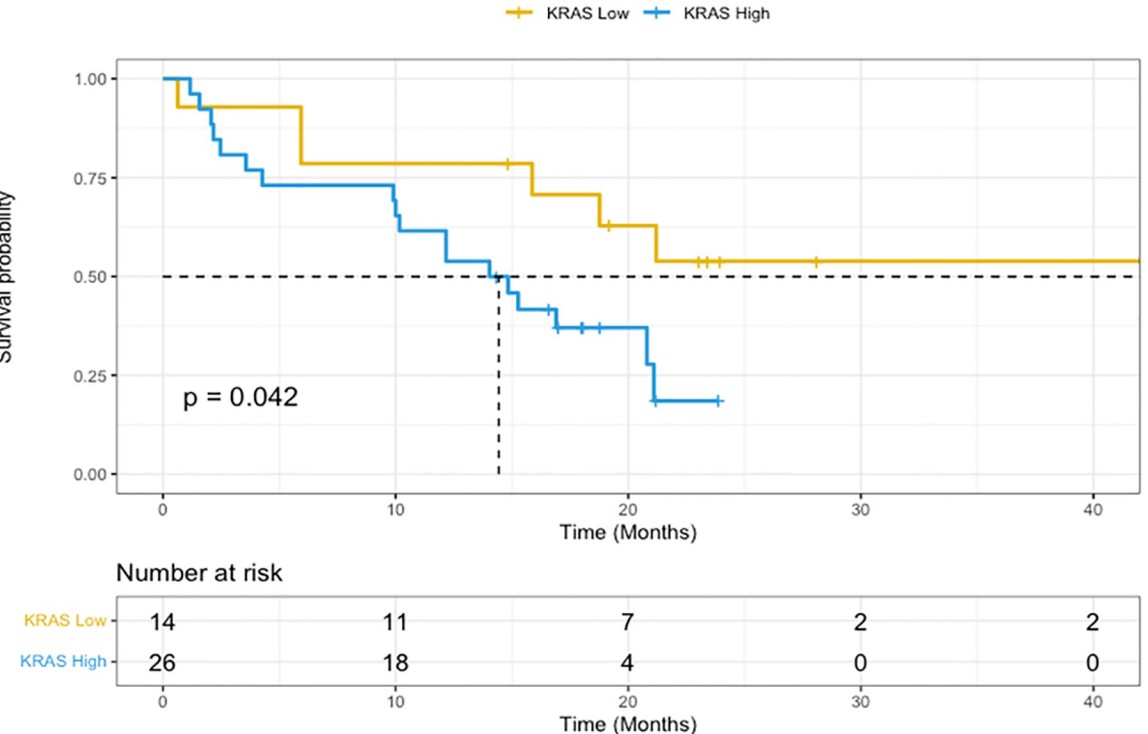

**Fig 3. Overall survival of RAS-mutated metastatic colorectal cancer based on allele frequency.** Kaplan-Meier curve comparing KRAS high variant allele frequent as defined by over 20% (yellow) vs KRAS low variant allele frequent as defined by below 20% (blue). Dotted line represents median (12.1 months versus not reached, HR 2.2, 95%CI 1.3–6.7, p = 0.04).

NSCLC who were treated with EGFR targeting TKIs. This was also supported by Ding et al. who similarly demonstrated a possible prognostic role of allele frequency in EGFR-mutated NSCLC [6]. Furthermore, BRAF allele frequency has been shown to be of prognostic relevance in melanoma [10].

To our knowledge, our study is the first to suggest that NGS may provide clinically relevant information for prognostication of patients with metastatic CRC and high RAS allele mutation frequency. Our data suggests that patients with high RAS allele mutation frequency trended towards reduced overall survival as measured from diagnosis of metastatic disease. This raises questions in clinical decision making. Should clinicians be referring patients with high RAS allele frequency for clinical trials given poor prognosis or is there a role for increased surveillance imaging in these patients to detect earlier treatment failure? Our data set did not

**Table 5. Genetic variations based on region of birth in colorectal cancer.**

| Region of Birth | N | KRAS | BRAF | TP53 | PIK3CA | MET | HER2 |
|---|---|---|---|---|---|---|---|
| Australia / New Zealand | 112 | 42 (38%) | 23 (21%) | 37 (33%) | 16 (14%) | 6 (5%) | 3 (3%) |
| South East Asia | 23 | 11 (48%) | 3 (13%) | 6 (26%) | 4 (17%) | 1 (4%) | 0 (0%) |
| Pacific Islands | 2 | 1 (50%) | 0 (0%) | 2 (100%) | 0 (0%) | 0 (0%) | 0 (0%) |
| Eastern Europe | 28 | 11 (40%) | 0 (0%) | 8 (29%) | 3 (11%) | 1 (4%) | 1 (4%) |
| Middle East | 11 | 4 (36%) | 2 (18%) | 5 (45%) | 2 (18%) | 0 (0%) | 0 (0%) |
| Other | 4 | 2 (50%) | 1 (25%) | 2 (50%) | 0 (0%) | 0 (0%) | 0 (0%) |

demonstrate a lack of upfront response to chemotherapy but further, larger studies are required to answer these questions. These same questions can be posed for patients with CRC tissue that carries the combination of *TP53* and *RAS* mutation who show worse prognosis in our cohort (Fig 2). Only recently has the combined presence of both *TP53* and *RAS* mutation been shown to be associated with reduced survival outcomes and our NGS data adds to the existing literature [11]. This data would not be available using more limited traditional mutation testing methods of immunohistochemistry or PCR thereby demonstrating the benefit of routine targeted NGS in metastatic CRC. Recently, early data has also proposed prognostic utility of BRAF allele frequency in metastatic CRC [12], thus further demonstrating the potential utility of routine NGS testing. Our data did not demonstrate any differences in survival based on *BRAF* allele frequency but findings may be limited to small numbers of BRAF-V600E mutated CRC in our cohort.

Our retrospective cohort study demonstrates the utility of routine NGS testing in stage IV CRC patients and describes the tumour mutational landscape in the Australian population. NGS is currently recommended as an alternative to mutation testing by PCR in CRC. Although the use of multigene sequencing panels is recommended in large research centres in the context of screening for access to novel potentially therapeutic targets, this needs to be balanced with cost and the detection of mutations of unknown clinical significance. In Table 4 we have listed examples of clinical trials that have been or are being conducted to investigate the role of these mutations either specifically in CRC or in all solid organ malignancies. All of the trials in this table are early phase I to II studies. The results of these studies may shift these mutations from potential tier 3 to 4 to a higher tier if clinical utility is determined but may also downstage these variants to Tier 5 if they are determined to be bystander mutations. Hence, we can see that extended panel testing currently remains investigational but is critical in a research or clinical trial context in identifying genes that are potential targets for directed therapy. It is unclear if larger gene panels may identify more genes with actionable targets in early phase studies. Furthermore, identification of resistance genes and those that confer worse prognosis is possible by targeted NGS. Ongoing studies are required to further clarify the role of specific mutations as either therapeutic targets, resistance pathways or prognostic markers. In addition, specific NGS panels for individual cancers is also important to consider should NGS testing be incorporated into routine practice. Of note, APC (Adenomatous Polyposis Coli) is recommended as part of young CRC and hereditary CRC work-ups but is **not** included as part of the Oncomine panel. This is a limitation with pan-cancer NGS testing and more targeted selection is required to ensure optimal testing and resource allocation.

In the future, NGS testing is likely to become increasingly common in the diagnostic setting given the increased therapeutic targets in cancer treatment. Although NGS provides a wealth of biometric data, the clinical utility of much of this information is currently unknown. Other potential challenges of routine NGS testing is a lack of standardization across testing centres and funding for NGS testing. NGS testing costs can range into thousands of dollars depending on the number of genes tested and whether both DNA and RNA are sequenced.

## 5. Conclusion

Our study adds to the literature by describing the tumour mutational landscape of patients with stage IV CRC in a diverse Australian population. In addition to identifying patients with genomic alterations suitable for clinically proven therapeutic options, standard of care NGS testing revealed an additional 22% of cases with Tier II & III ESCAT mutations in CRC. 43% of patients also demonstrated potentially targetable Tier III & IV mutations. Our data also highlighted that patients with concurrent TP53 and RAS mutations have significant worse

prognosis than those with RAS mutation alone. Furthermore, we were able to demonstrate that allele frequency has a potential clinical role in prognostication as well as therapeutic decision making. This supports the routine use of extended gene panels in identifying potentially actionable mutations in standard clinical practice. Future research is directed towards investigating the utility and cost effectiveness of NGS panels, especially with the advent of larger multigene panels, as well as standardisation of NGS testing in metastatic CRC.

## Supporting information

**S1 Data.**
(XLSX)

## Author Contributions

**Conceptualization:** Udit Nindra, Abhijit Pal, Tara L. Roberts, Therese M. Becker, Tristan Rutland, Weng Ng, C. Soon Lee, Wei Chua.

**Data curation:** Udit Nindra, Kate Wilkinson, Mahtab Farzin, Tristan Rutland.

**Formal analysis:** Udit Nindra, Wei Chua.

**Investigation:** Udit Nindra.

**Methodology:** Udit Nindra, Kate Wilkinson, Wei Chua.

**Project administration:** Udit Nindra.

**Resources:** Udit Nindra.

**Supervision:** Abhijit Pal, Vivienne Lea, Stephanie Hui-Su Lim, Kate Wilkinson, Ray Asghari, Tara L. Roberts, Therese M. Becker, Mahtab Farzin, Tristan Rutland, Mark Lee, Scott MacKenzie, Weng Ng, Bin Wang, C. Soon Lee, Wei Chua.

**Validation:** Udit Nindra, Abhijit Pal, Kate Wilkinson, Therese M. Becker, Weng Ng, C. Soon Lee, Wei Chua.

**Visualization:** Udit Nindra, Vivienne Lea, Stephanie Hui-Su Lim, Kate Wilkinson, Tara L. Roberts, Weng Ng, C. Soon Lee, Wei Chua.

**Writing – original draft:** Udit Nindra, Therese M. Becker, Mahtab Farzin, Tristan Rutland, Weng Ng, C. Soon Lee, Wei Chua.

**Writing – review & editing:** Udit Nindra, Abhijit Pal, Vivienne Lea, Stephanie Hui-Su Lim, Kate Wilkinson, Ray Asghari, Therese M. Becker, Mark Lee, Scott MacKenzie, Weng Ng, Bin Wang, C. Soon Lee, Wei Chua.

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
