## [Decision Letter · Decision Letter 0]

18 Jul 2023

PONE-D-23-14740Multigene panel next generation sequencing in metastatic colorectal cancer in an Australian populationPLOS ONE

Dear Dr. Nindra,

Thank you for submitting your manuscript to PLOS ONE. After careful consideration, we feel that it has merit but does not fully meet PLOS ONE’s publication criteria as it currently stands. Therefore, we invite you to submit a revised version of the manuscript that addresses the points raised during the review process. Please submit your revised manuscript by Sep 01 2023 11:59PM. If you will need more time than this to complete your revisions, please reply to this message or contact the journal office at plosone@plos.org. Please include the following items when submitting your revised manuscript:A rebuttal letter that responds to each point raised by the academic editor and reviewer(s). You should upload this letter as a separate file labeled 'Response to Reviewers'.A marked-up copy of your manuscript that highlights changes made to the original version. You should upload this as a separate file labeled 'Revised Manuscript with Track Changes'.An unmarked version of your revised paper without tracked changes. You should upload this as a separate file labeled 'Manuscript'.

We look forward to receiving your revised manuscript.

Kind regards,

Avaniyapuram Kannan Murugan, M.Phil., Ph.D.

Academic Editor

PLOS ONE

Journal Requirements:

Reviewers' comments:

Reviewer's Responses to Questions

**Comments to the Author**

1. Is the manuscript technically sound, and do the data support the conclusions?

Reviewer #1: Yes

Reviewer #2: Yes

2. Has the statistical analysis been performed appropriately and rigorously? 

Reviewer #1: Yes

Reviewer #2: Yes

3. Have the authors made all data underlying the findings in their manuscript fully available?

Reviewer #1: Yes

Reviewer #2: No

4. Is the manuscript presented in an intelligible fashion and written in standard English?

Reviewer #1: Yes

Reviewer #2: Yes

5. Review Comments to the Author

Reviewer #1: The authors present a mutational landscape analysis of metastatic CRC patients using a targeted sequencing panel. The results suggest the importance of RAS /RAS-P53 mutations on survival in stage IV patients. More importantly, this analysis shows the value of NGS sequencing panels for stratifying patients into clinically actionable sub-groups.

1. Overall findings:

The work focuses on the importance of mutational profiling of patients to determine clinically actionable groups. It uses the survival dependence of KRAS-P53 mutations in an ethnically diverse cohort to show the utility of targeted NGS panels. It is important to expose this work and present the analytical strategy for other researchers/clinicians to consider. This multi-ethnic cohort represents the addition of new/novel data to the field.

The data/analysis presented does support the conclusions from the paper. The sample sizes could be larger, but this is frequently observed with human subject data. Generally, the manuscript lacks some key and relevant references, and discussion of the trends in context of the current similar findings in the field. For example, the KRAS-P53 effect on survival is documented in CRC and other cancers. The analysis results could be probed in more detail and it is not stated whether this is the only mutation pattern that bore any effect on survival trends.

2. Statistical analysis:

The sample size is small for statistical validity using Kaplain Meier statistics. For example, 14 patients with both RAS-P53 mutations in Fig 2a. However CI values are given which helps support the analysis trend. This trend is not novel and has been observed in multiple other studies/spontaneous CRC subtypes. This should be discussed in the manuscript.

See:

https://journals.lww.com/amjclinicaloncology/fulltext/2004/02000/simultaneous_mutations_in_k_ras_and_tp53_are.8.aspx and https://pubmed.ncbi.nlm.nih.gov/33351476/

Were these the only patterns identified that correlated with survival from the current study/oncomine profiles ?

What was the breakdown of sex/ethnic groupings in the KRAS-P53 mutated sets ? vs random sampling.

Other technical points include sharing a code/script/repo for the analysis code.

3. Legibility:

The manuscript was well written and easy to understand. The study goals were well described and the impact of the work clearly put into context with the current field.

4. Data Availability:

The research involves human subjects and consequently is exempt from the PLoS rules around making all datasets publicly accessible. The authors suggest that these data could be available on request. It is unclear why the data from this study might not be deposited in a protected-data site like dbGap or BioProject archive.

Reviewer #2: The manuscript entitled “Multigene panel next generation sequencing in metastatic colorectal cancer in an Australian population” by Nindra et al. reported a 50-gene panel analysis of a large cohort of CRC samples by next generation sequencing (NSG) and concluded this approach is beneficial as it is likely to identify actionable genomic alterations. However, additional analyses are needed to support the conclusions.

1. APC is the most frequently mutated gene in colon cancer, but it is not included in the panel. Do the authors have any information about APC mutational status of these samples? If yes, taking into account the APC mutations, whether the conclusion is different from the current one?

2. Is there any correlation of mutation loads/amount of mutated genes detected with the MMR status in the samples tested?

3. Whether there are any correlation of the different mutations with the responses to any therapies the patients received?

4. What kinds of mutations were detected by this method? Do these mutations include non-sense mutations or only include the ones that have been reported to alter the function of the genes? If non-sense mutations were included, the authors should consider separate them from the other damaging mutations.

5. Related to the Introduction section, as far as I know, BRAF is a S/T kinase, and therefore the encorafenib is not a tyrosine kinase inhibitor.

6. It should be CDKN2A instead of CDK2NA in Results 3.1.1. Also, it is confusing that the authors claimed there was no ongoing trials testing the roles of the 12 genes. At least inhibitors for MTOR and KIT are readily available.

7. Whether the difference of RAS mutations among diverse ethnic groups is statistically significant?

6. PLOS authors have the option to publish the peer review history of their article (what does this mean?). If published, this will include your full peer review and any attached files.

Reviewer #1: **Yes: **Anna Lobley

Reviewer #2: No

---

## [Author Response · Author response to Decision Letter 0]

20 Jul 2023

Response to Reviewers:

Comment 1:

The authors present a mutational landscape analysis of metastatic CRC patients using a targeted sequencing panel. The results suggest the importance of RAS /RAS-P53 mutations on survival in stage IV patients. More importantly, this analysis shows the value of NGS sequencing panels for stratifying patients into clinically actionable sub-groups.

Answer 1:

We thank the reviewer for this positive feedback. There is a paucity of data in the Australian context on NGS testing in the colorectal space and thus our manuscript is novel in adding to that information. We feel that our cohort has a significant degree of multi-cultural variability, reflective of the modern Australian cohort, and thus is a relevant literature manuscript

Comment 2:

The work focuses on the importance of mutational profiling of patients to determine clinically actionable groups. It uses the survival dependence of KRAS-P53 mutations in an ethnically diverse cohort to show the utility of targeted NGS panels. It is important to expose this work and present the analytical strategy for other researchers/clinicians to consider. This multi-ethnic cohort represents the addition of new/novel data to the field.

Answer 2:

Again, we thank the reviewer for this positive feedback

Comment 3:

The data/analysis presented does support the conclusions from the paper. The sample sizes could be larger, but this is frequently observed with human subject data. Generally, the manuscript lacks some key and relevant references, and discussion of the trends in context of the current similar findings in the field. For example, the KRAS-P53 effect on survival is documented in CRC and other cancers. The analysis results could be probed in more detail and it is not stated whether this is the only mutation pattern that bore any effect on survival trends.

Answer 3:

Again, we thank the reviewer for this positive feedback. This cohort of 180 patients is one of the largest NGS testing samples of CRC done in Australia. We did not identify any other mutation patterns that affected survival. Of note, PIC3KA correlated with no difference in clinical outcomes and this has been mentioned in our manuscript.

Comment 4:

The sample size is small for statistical validity using Kaplain Meier statistics. For example, 14 patients with both RAS-P53 mutations in Fig 2a. However, CI values are given which helps support the analysis trend. This trend is not novel and has been observed in multiple other studies/spontaneous CRC subtypes. This should be discussed in the manuscript.

See:

https://journals.lww.com/amjclinicaloncology/fulltext/2004/02000/simultaneous_mutations_in_k_ras_and_tp53_are.8.aspx and https://pubmed.ncbi.nlm.nih.gov/33351476/

Answer 3:

Again, we thank the reviewer for this positive feedback. The reference “https://pubmed.ncbi.nlm.nih.gov/33351476/” mentioned is already cited in our discussion and we have made reference to previous manuscripts highlighint the poor prognostic trnd in RAS and P53 mutated CRC. 

We did not include “https://journals.lww.com/amjclinicaloncology/fulltext/2004/02000/ simultaneous_mutations_in_k_ras_and_tp53_are.8.aspx” as the results are from 2004 and treatment outcomes from that period are dated and hard to correlate with current standard of care regimens.

Comment 4:

Were these the only patterns identified that correlated with survival from the current study/oncomine profiles ?

Answer 4:

We did not identify any other mutation patterns that affected survival. Of note, PIC3KA correlated with no difference in clinical outcomes and this has been mentioned in our manuscript.

Comment 5:

What was the breakdown of sex/ethnic groupings in the KRAS-P53 mutated sets ? vs random sampling.

Answer 5:

There was no grouping or random sampling applied in the cohort. We used 180 sequential samples of CRC.

Comment 6:

The manuscript was well written and easy to understand. The study goals were well described and the impact of the work clearly put into context with the current field. 

Answer 6:

Again, we thank the reviewer for this positive feedback

Comment 7:

The research involves human subjects and consequently is exempt from the PLoS rules around making all datasets publicly accessible. The authors suggest that these data could be available on request. It is unclear why the data from this study might not be deposited in a protected-data site like dbGap or BioProject archive.

Answer 7:

We have uploaded a de-identified minimum data set as part of the supplemental material

Comment 8:

APC is the most frequently mutated gene in colon cancer, but it is not included in the panel. Do the authors have any information about APC mutational status of these samples? If yes, taking into account the APC mutations, whether the conclusion is different from the current one?

Answer 8:

APC genes are not mutated in colon cancer but are germline mutations that predispose to increased risk of colon cancer. NGS testing is done on somatic tumour tissue to look for tumour specific rather than germline specific mutations. Thus this comment is not relevant to our project with assesses the role of NGS testing in somatic tissue mutation identification in CRC.

Comment 9:

Is there any correlation of mutation loads/amount of mutated genes detected with the MMR status in the samples tested?

Answer 9:

No specific correlation is noted between tumour mutational burden and MMR – although this is not a key focus or specific to NGS testing and thus outside the scope of our paper

Comment 10:

Whether there are any correlation of the different mutations with the responses to any therapies the patients received?

Answer 10:

Upfront response rate to first line chemotherapy was not different between mutations but given the retrospective nature of the project this is a difficult question to answer. Prospective cohort studies are required to accurately assess tumour response using RECIST1.1 criteria to assess this and cannot be inferenced from our data.

Comment 11:

What kinds of mutations were detected by this method? Do these mutations include non-sense mutations or only include the ones that have been reported to alter the function of the genes? If non-sense mutations were included, the authors should consider separate them from the other damaging mutations.

Answer 11:

We have described the types and nature of mutation testing in our manuscript as follows: -“Next generation sequencing was performed with the Ion Torrent Genexus integrated next generation sequencer using the Oncomine Precision AssayTM, which is 50-gene NGS panel that was introduced to our laboratory in June 2021. Only DNA hotspot mutations and copy number variations (CNVs) were assessed. RNA sequencing for fusions was not performed. The Ion Torrent Genexus System fully automates the specimen-to-report workflow. It automates all steps of the targeted NGS workflow starting from extraction, purification and quantification of nucleic acid. The Genexus Purification System replaces sample prep by extracting and quantifying nucleic acids. This is followed by automation of the library preparation (including DNA synthesis), template preparation and sequencing. Lastly, primary data analysis is also automated and variant reporting for DNA applications are performed using the OncomineTM Reporter software, a Thermo Fisher supplied bioinformatics analytic tools. The reported results from the 50-gene OncomineTM Precision Assay are filtered to provide a summary of the variants and copy number variation as well as the variant allele frequency.”

Comment 12:

Related to the Introduction section, as far as I know, BRAF is a S/T kinase, and therefore the encorafenib is not a tyrosine kinase inhibitor.

Answer 12:

Amended; Encorafenib acts as an ATP-competitive RAF kinase inhibitor – we have updated the manuscript to reflect this.

Comment 13:

It should be CDKN2A instead of CDK2NA in Results 3.1.1.

Answer 13:

Amended; Thank you.

Comment 14:

Whether the difference of RAS mutations among diverse ethnic groups is statistically significant?

Answer 14:

There was a slightly higher proportion of RAS mutations in patients born in Australia compared with overseas but the results were not statistically significant

Comment 15:

PLOS authors have the option to publish the peer review history of their article (what does this mean?). If published, this will include your full peer review and any attached files.

Answer 15:

Yes, happy for publication of peer review history

---

## [Decision Letter · Decision Letter 1]

3 Sep 2023

PONE-D-23-14740R1Multigene panel next generation sequencing in metastatic colorectal cancer in an Australian populationPLOS ONE

Dear Dr. Nindra,

Thank you for submitting your manuscript to PLOS ONE. After careful consideration, we feel that it has merit but does not fully meet PLOS ONE’s publication criteria as it currently stands. Therefore, we invite you to submit a revised version of the manuscript that addresses the points raised during the review process.

We look forward to receiving your revised manuscript.

Kind regards,

Avaniyapuram Kannan Murugan, M.Phil., Ph.D.

Academic Editor

PLOS ONE

Journal Requirements:

Reviewers' comments:

Reviewer's Responses to Questions

**Comments to the Author**

1. If the authors have adequately addressed your comments raised in a previous round of review and you feel that this manuscript is now acceptable for publication, you may indicate that here to bypass the “Comments to the Author” section, enter your conflict of interest statement in the “Confidential to Editor” section, and submit your "Accept" recommendation.

Reviewer #1: All comments have been addressed

Reviewer #2: (No Response)

2. Is the manuscript technically sound, and do the data support the conclusions?

Reviewer #1: Yes

Reviewer #2: Yes

3. Has the statistical analysis been performed appropriately and rigorously? 

Reviewer #1: Yes

Reviewer #2: I Don't Know

4. Have the authors made all data underlying the findings in their manuscript fully available?

Reviewer #1: No

Reviewer #2: Yes

5. Is the manuscript presented in an intelligible fashion and written in standard English?

Reviewer #1: Yes

Reviewer #2: Yes

6. Review Comments to the Author

Reviewer #1: The authors have addressed the outstanding queries in their responses, however, I'm not sure the minimal data added as supplementary materials is sufficient to support the findings in the manuscript. Were there any sample qc results for the patients samples ie. Depth and read numbers or similar ? The only other query I have is that the confidence intervals appear to be shown only for one of the groups on the test... could the authors clarify if the baseline kras data a single sample or the CI interval not shown ? Following clarity in this aspect I would recommend the work be published.

Reviewer #2: In this recently revised manuscript, the authors have made significant strides in addressing the concerns I had previously raised. However, there remains an issue pertaining to APC mutations that I strongly disagree with the authors on. In their rebuttal letter, the authors assert that APC is exclusively subject to germline mutations and not somatic mutations in the context of colorectal cancer (CRC). I find this assertion problematic due to the well-established knowledge spanning over two decades, which affirms that somatic APC mutations are indeed among the earliest events in the development of non-hereditary colorectal carcinogenesis. These somatic mutations are pivotal in driving the formation of early adenoma/dysplastic crypt structures.

While it is reasonable to acknowledge that the commercial 50-gene panel used in the Oncomine Precision AssayTM might not have initially incorporated APC, I believe that as professionals and investigators focused on CRC, the authors should have at least initiated a scientific discussion concerning the potential limitations of omitting this widely recognized driver gene from their study. It seems unlikely that the authors were unaware of prior genetic research pertaining to APC's role in CRC.

With the exception of this concern, I do not have any further queries or reservations regarding the content of this manuscript.

7. PLOS authors have the option to publish the peer review history of their article (what does this mean?). If published, this will include your full peer review and any attached files.

Reviewer #1: No

Reviewer #2: No

---

## [Author Response · Author response to Decision Letter 1]

6 Sep 2023

Response to Reviewers

Comment 1:

The authors have addressed the outstanding queries in their responses, however, I'm not sure the minimal data added as supplementary materials is sufficient to support the findings in the manuscript. Were there any sample qc results for the patients samples ie. Depth and read numbers or similar? 

Answer 1:

Depth and read numbers were not available on next generation sequencing reports. For example, I have attached a redacted copy of one example of the output next generation sequencing obtained. As you can see below, such information is not present.

Comment 2:

The only other query I have is that the confidence intervals appear to be shown only for one of the groups on the test... could the authors clarify if the baseline kras data a single sample or the CI interval not shown ? 

Answer 2:

Apologies, that is an omission on our end. The confidence intervals are now included in the main text and figure.

Comment 3:

Following clarity in this aspect I would recommend the work be published.

Answer 3:

We thank the authors for their feedback and hope PLOS ONE agrees for publication.

Comment 4: In this recently revised manuscript, the authors have made significant strides in addressing the concerns I had previously raised. However, there remains an issue pertaining to APC mutations that I strongly disagree with the authors on. In their rebuttal letter, the authors assert that APC is exclusively subject to germline mutations and not somatic mutations in the context of colorectal cancer (CRC). I find this assertion problematic due to the well-established knowledge spanning over two decades, which affirms that somatic APC mutations are indeed among the earliest events in the development of non-hereditary colorectal carcinogenesis. These somatic mutations are pivotal in driving the formation of early adenoma/dysplastic crypt structures. While it is reasonable to acknowledge that the commercial 50-gene panel used in the Oncomine Precision AssayTM might not have initially incorporated APC, I believe that as professionals and investigators focused on CRC, the authors should have at least initiated a scientific discussion concerning the potential limitations of omitting this widely recognized driver gene from their study. It seems unlikely that the authors were unaware of prior genetic research pertaining to APC's role in CRC. With the exception of this concern, I do not have any further queries or reservations regarding the content of this manuscript.

Answer 4:

We thank the authors for their feedback. We agree with the reviewer that somatic APC mutations are among the earliest in the development of colorectal cancer. APC mutations are typically screened using specific PCR or NGS in certain subgroups.

Based on Australian guidelines, testing for APC using NGS is only indicated in patients who have isolated colorectal cancer under the age of 40, those with KRAS G12C mutations or those with family histories suggestive of hereditary cancer. This gene testing is not part of the Oncomine 50-gene panel, which in itself if an investigational tool and not part of standard government funded practice in Australia.

We have added a discussion regarding the APC gene in our revised manuscript, but it’s relevance and utility is outside the scope of our manuscript.

---

## [Editor Report · Decision Letter 2]

12 Sep 2023

Multigene panel next generation sequencing in metastatic colorectal cancer in an Australian population

PONE-D-23-14740R2

Dear Dr. Nindra,

We’re pleased to inform you that your manuscript has been judged scientifically suitable for publication and will be formally accepted for publication once it meets all outstanding technical requirements.

Kind regards,

Avaniyapuram Kannan Murugan, M.Phil., Ph.D.

Academic Editor

PLOS ONE
---

## [Editor Report · Acceptance letter]

19 Sep 2023

PONE-D-23-14740R2 

Multigene panel next generation sequencing in metastatic colorectal cancer in an Australian population 

Dear Dr. Nindra:

I'm pleased to inform you that your manuscript has been deemed suitable for publication in PLOS ONE. Congratulations! Your manuscript is now with our production department. 

Kind regards, 

on behalf of

Dr. Avaniyapuram Kannan Murugan 

Academic Editor

PLOS ONE